# Intensive Rehabilitation Program in Older Adults with Stroke: Therapy Content and Feasibility—Preliminary Results from the BRAIN-CONNECTS Study

**DOI:** 10.3390/ijerph20064696

**Published:** 2023-03-07

**Authors:** Andrea Morgado-Pérez, Maria Coll-Molinos, Ruben Valero, Miriam Llobet, Nohora Rueda, Andrea Martínez, Sonia Nieto, Cindry Ramírez-Fuentes, Dolores Sánchez-Rodríguez, Ester Marco, Josep Puig, Esther Duarte

**Affiliations:** 1Rehabilitation Research Group, Hospital del Mar Research Institute, Dr. Aiguader, 88, 08003 Barcelona, Catalonia, Spain; 2Physical Medicine and Rehabilitation Department, Parc de Salut Mar (Hospital de l’Esperança), Sant Josep de la Muntanya 12, 08024 Barcelona, Catalonia, Spain; 3Department of Medicine, Universitat Autònoma de Barcelona, Passeig de la Vall d’Hebron, 119-129, 08035 Barcelona, Catalonia, Spain; 4Geriatrics Department, Brugmann University Hospital, Université Libre de Bruxelles, Place A. Van Gehuchten 4, 1020 Brussels, Belgium; 5WHO Collaborating Centre for Public Health Aspects of Musculo-Skeletal Health and Ageing, Division of Public Health, Epidemiology and Health Economics, University of Liège, 4000 Liège, Belgium; 6Faculty of Health and Life Sciences, Universitat Pompeu Fabra, Aiguader 80, 08003 Barcelona, Catalonia, Spain; 7Department of Radiology, Biomedical Research Institute Imaging Research Unit, Diagnostic Imaging Institute, Doctor Josep Trueta University Hospital of Girona, Avinguda de França, s/n, 17007 Girona, Catalonia, Spain

**Keywords:** feasibility, stroke, intensive rehabilitation program, older adults

## Abstract

The main objective was to assess the feasibility of an intensive rehabilitation program (IRP) for stroke patients; and secondly, to detect eventual age-related differences in content, duration, tolerability, and safety in a prospective observational cohort of patients diagnosed with subacute stroke, admitted to inpatient rehabilitation (BRAIN-CONNECTS project). Activities during physical, occupational and speech therapy, and time dedicated to each one were recorded. Forty-five subjects (63.0 years, 77.8% men) were included. The mean time of therapy was 173.8 (SD 31.5) minutes per day. The only age-related differences when comparing patients ≥65 and <65 years were a shorter time allocated for occupational therapy (−7.5 min (95% CI −12.5 to −2.6), *p* = 0.004) and a greater need of speech therapy (90% vs. 44%) in the older adults. Gait training, movement patterns of upper limbs, and lingual praxis were the most commonly performed activities. Regarding tolerability and safety, there were no losses to follow-up, and the attendance ratio was above 95%. No adverse events occurred during any session in all patients. Conclusion: IRP is a feasible intervention in patients with subacute stroke, regardless of age, and there are no relevant differences on content or duration of therapy.

## 1. Introduction

There are more than 101 million stroke survivors in the world, and every year, more than 12.2 million individuals suffer a new stroke [1]. As the population increases and lives longer, the incidence of stroke, long-term sequelae, and associated costs are expected to grow dramatically; about one of every four people over the age of 25 will have a stroke in their lifetime [1]. Many survivors experience motor, sensory, perceptual, and cognitive impairments, and require rehabilitation in the months following the stroke [2].

The intensive rehabilitation programs (IRP) offer multidisciplinary care for patients with subacute stroke, and lead to better functional results and less institutionalization. The individual’s cognitive status, comorbidity, and previous functional level are determining factors for admitting patients to IRP [3]. Advanced age is considered a risk of receiving poorer quality of care and a limitation for rehabilitation outcomes following stroke [4]. The influence of age on stroke rehabilitation has been a controversial issue, with studies showing a negative effect on outcomes, whereas others have not found any relationship. Although older patients present more dependence in activities of daily living (ADL) three months after stroke, no significant differences in the efficacy of the rehabilitation have been found in older stroke survivors [5]. Some quality care indicators such as the number of computerized tomography head scans and carotid image analyses in older patients with stroke seem to be lower than in younger adults [4], but there is presently no evidence showing that content and/or intensity of rehabilitation therapies are different in the oldest patients with stroke [6].

Current guidelines recommend that rehabilitation programs should provide at least 3 h per day of physiotherapy, occupational therapy, and speech therapy, 5 days a week [7,8]. Although age in itself is not considered as a selection criterion for IRP, in daily practice, older patients, who might have higher medical and social needs, are often referred to nursing homes and intermediate care settings where rehabilitation programs are of lower intensity.

The medical literature on stroke IRP does not usually report the intensity and content of interventions. Factors such as patient tolerance, the resources of each facility, and the physical environment can modify the intensity of the given therapies [3,7,9,10,11,12]. The measurement of the interventions included in the rehabilitation programs represents a challenge for professionals, and it is essential to advance in the demonstration of their benefits. In daily practice, the rehabilitation process starts with a clinical and functional assessment of patients, followed by the establishment of objectives according to individual needs and a therapeutic plan to achieve them. However, the exact content and description of the interventions are usually unknown. The complex variety of treatment goals and therapies have been called the “black box” of rehabilitation [10,11,12]. There is a need for an adequate system to classify the wide range of interventions [13]. Moreover, the standardization of the therapies will provide the external validity necessary for clinical trials. Several objective systems to measure the activities and the time dedicated to each of them have been proposed [12,14,15], but only one study gives detailed information about physical, occupational, and speech therapies [3]. Feasibility studies are important, especially in the development of complex interventions and multidisciplinary programs [10]. Before testing the efficacy of an intervention through a clinical trial, it is recommended to verify that it can be carried out as proposed in clinical settings [16].

Based on these considerations, this study aimed at assessing the feasibility of an IRP for stroke patients by recording the type of activities and quantifying the time dedicated to each one during physiotherapy, occupational, and speech therapy; and secondly, to assess eventual differences according to age.

## 2. Materials and Methods

### 2.1. Study Design

This study was a prospective observational cohort of patients diagnosed with subacute stroke, admitted to inpatient rehabilitation from January 2020 to June 2022. The cohort was part of the BRAIN-CONNECTS project, a multicenter prospective study that aims to determine the predictive value of the functional magnetic resonance analysis of brain connectivity in predicting functional outcomes in the rehabilitation of patients diagnosed with subacute stroke. This study was approved by the local committee of the Hospital del Mar Medical Research Institute (Barcelona, Spain; Project ID: 34/C/2017); all participants signed an informed consent.

### 2.2. Study Setting

This study was carried out in the Physical Medicine and Rehabilitation Department of a tertiary hospital accredited by the European Stroke Organization as a specialized center for the intensive rehabilitation for patients diagnosed with acute and subacute stroke. The center offers intensive inpatient and outpatient rehabilitation programs for patients with a good recovery potential according to the following criteria: absence of severe cognitive impairments (Montreal Cognitive Assessment (MoCA) ≥ 20), low comorbidity (Charlson Index < 3), and previous functional independence (Modified Rankin Scale (mRS) ≤ 2); with age and social situation not being determining factors for admission. The intensive rehabilitation ward has eighteen beds, three specialists in Physical Medicine and Rehabilitation, four physiotherapists, two occupational therapists, one speech therapist, one social worker, and one neuropsychologist, as well as a team of nurses and nursing assistants trained in neurological rehabilitation.

### 2.3. Participants

Patients with subacute stroke admitted for intensive rehabilitation were eligible for the BRAIN-Connects study if they fulfilled the following criteria: 18 years or older, first intracerebral ischemic or hemorrhagic stroke confirmed by neuroimaging, less than 3 weeks after the stroke onset, moderate-to-severe neurological impairment (National Institute of Health Stroke Scale [NIHSS] between 4 and 13), and the absence of a language barrier. Patients with any other neurological or psychiatric condition were excluded. 

### 2.4. Intensive Rehabilitation Program

The IRP aims to achieve the highest possible level of functionality and autonomy in basic ADL, as well as to facilitate the transition from hospital to home and social reintegration. The program follows a patient-centered process with cyclical stages that include: an assessment of individual needs, the setting of goals, therapeutic interventions, and re-assessment. Patients and caregivers actively participate in all the stages of the process.

In the initial neurological and functional assessment, patients, caregivers, and professionals participate in the establishment of the therapeutic goals. Physical, occupational, and speech therapists assess the patient’s mobility, performance in ADL, and communication/swallowing, respectively. In the first week of admission, the cognitive functions and the emotional state are assessed by a neuropsychologist. The multidisciplinary team meets weekly to review the patient’s progress and re-define specific goals.

The program consists of at least three scheduled hours of physical, occupational, and speech therapies per day, at least 5 days per week according to individual needs. The physiotherapy sessions address global mobility, balance, transfers, and gait. The occupational therapy sessions focus on training bodily functions such as movement, sensation, perception, and cognition, as well as ADL. Speech therapy sessions focus on managing dysphagia, improving language skills, and recovering from motor speech disorders. The therapy sessions are spread over the day to ensure the rest and active participation of patients. Although visits are usually allowed, in the period in which this study was carried out, most of the patients only had telephone or videoconference contact with their family and friends due to the restrictions caused by the SARS-CoV-2 pandemic.

During the hospital stay, patients, family members, and/or caregivers are invited to attend a one-hour educational session led by an occupational therapist. This session aims to offer information and training on how to manage the consequences of stroke and how to provide discharge support resources for patients and caregivers. This intervention has been demonstrated to improve knowledge about rehabilitation and the satisfaction of patients and caregivers with the rehabilitation program for stroke [17].

### 2.5. Treatment Record

At the end of each session, the therapist responsible for each patient recorded the number of minutes dedicated to each activity, the duration of sessions, and eventual adverse events, by using the “Data Collection Notebook” (Appendix A). The record of every possible activity within each therapy was created based on the proposals of other authors [12,14,15], and selected by the members of the rehabilitation multidisciplinary team according to the center availability (Appendix A). 

### 2.6. Study Endpoints

The main study endpoints were to describe the content and feasibility of the rehabilitation program. The content was defined by the sum of activities performed in each therapy, as described in Table 1. Therapy requirements, days of therapy, and time spent on daily sessions were recorded, as well as the number of patients performing each activity.

Patients were categorized according to the daily therapy dose: less or more than 180 min per day, as recommended in the current guidelines [7,8]. Feasibility was assessed by measuring tolerability and safety outcomes. Tolerability variables included: lost to follow-up, attendance (mean attended sessions) to calculate ratio number of sessions performed/programmed (%), reasons for not attendance (medical visits, clinical events, or other reasons), reasons for early termination in at least one session (e.g., high fatigability), and treatment interruption defined by missing 3 or more consecutive sessions due to medical visits and/or general discomfort. Safety variables included adverse events such as falls, musculoskeletal injuries, and others.

Baseline demographic and clinical data at admission in the intensive rehabilitation unit such as age, gender, years of schooling, dominance (right/left-handed), smoking history, alcoholic habits (standard drink units/day), and comorbidities were recorded. Stroke onset, type of stroke (ischemic/hemorrhagic), affected hemisphere (right/left), location (Oxford classification), etiology (TOAST classification), use of reperfusion treatment, NIHSS score, disability (mRS), and performance in ADLs (Barthel Index) were also recorded. 

### 2.7. Statistical Analysis

Data were reported as absolute and percentage values for categorical variables, and as means and standard deviation (SD) for quantitative measurements. The assumption of normality of continuous variables was analyzed by the Kolmogorov–Smirnoff test. Univariate analysis was performed using a chi-squared test for categorical variables, and a paired double-tailed Student’s t-test for continuous variables, reporting difference means with 95% confidence intervals (95% CI). Statistical significance was set at *p* < 0.05. All statistical tests were performed using SPSS version 20.0 (Chicago, IL, USA).

## 3. Results

Forty-five subjects (mean age 63.0 years (SD 11.2), 77.8% men) were included. The demographic and clinical characteristics of participants are described in Table 2. Patients over 65 years had fewer years of schooling (95% CI, *p* = 0.011), and more prevalence of hypertension (*p* = 0.035). Large-artery atherosclerosis was more frequent in older adults (50% vs. 24%), and undetermined strokes was more frequent in adults under 65 years of age (40% vs. 11.1%) (*p* = 0.004). Despite these differences, older patients did not have more severe strokes (NIHSS score), more disability (mRS), or worse performance in ADLs (Barthel Index).

Table 3 shows the type and duration of therapies performed during the stay in the intensive rehabilitation unit. The mean length of stay was 16.4 days (SD 8.9), with 13.2 (SD 7.4) days in which patients performed any type of therapy, and a mean time of therapy per day of 173.8 (SD 31.5) minutes. No significant differences were found when comparing older patients with the group of younger adults, with the exception that the time allocated for occupational therapy was a few minutes shorter in the older adults (mean difference −7.5 min (95% CI −12.5 to −2.6, *p* = 0.004)). The oldest patients also had more need of speech therapy (90% vs. 44%), but this had no repercussion on the daily time allocated to speech therapy.

Patients were categorized according to if they achieved the 180-min standard of therapy. As shown in Table 4, patients who performed more than 180 min of daily therapy had significantly longer hospital stays and, consequently, therapy days. It is noteworthy that no women met the standard of 180 min per day, with no found differences on years of education, comorbidities, stroke type, affected hemisphere, etiology, NIHSS, mRS, or Barthel Index at admission. 

The most performed activities in physiotherapy were “gait training”, followed by “balance training” and “active exercises of affected limbs”. In occupational therapy, the most performed activities were “movement patterns of upper limbs”, followed by “ADL training” and “stroke education”. In speech therapy, the most performed activities were “lingual praxis”, “oral production”, and “neuromuscular electrical stimulation for dysphagia”. Figure 1 illustrates the therapeutic activities in adults and older adults with stroke. Once again, no significant differences among age groups were found.

Table 5 shows the tolerability and safety outcomes during the rehabilitation session. No patient was lost to follow-up, and attendance, calculated as the ratio of performed/scheduled sessions, was above 95% in both groups. The main reason for missing a session was having to attend external medical visits and/or examinations, as patients in this study required a magnetic resonance imagery within the first three weeks after stroke. The reasons for requiring early termination in at least one session were fatigability; poor active participation; or others such as headache, need of rest, etc. Once again, the most frequent reason for treatment interruption (missing ≥ three consecutive sessions) was attendance to external medical visits and/or examinations. No adverse events occurred during any session in all age groups. The most performed activities in physiotherapy were gait training, followed by balance training.

## 4. Discussion

This study not only shows the feasibility of the IRP in adults and older adults with subacute stroke, but it also opens the black box in rehabilitation with a description of activities and the spent time on each of them. Clinical manifestations and functional repercussions of stroke are so varied that no one could expect that treatment would be the same for everyone. This complex intervention makes it more difficult to classify activities and quantify the amount of therapy.

Rehabilitation interventions are poorly described in research studies, and it is often limited to measure the time spent in each therapy without taking into account the type and amount of activity and tasks performed. To move beyond simple models of rehabilitation that indicate the type of therapies performed, some authors have proposed to provide information on the number of repetitions for each movement performed and/or time-units spent on each activity [12,14]. However, these methods have limited applicability: it is not always clear which movements must be taken into account, and the measurement of time is also a difficult challenge. Our study proposes to record the type of activities (physiotherapy, occupational, and speech therapy), and to quantify the time spent on each of them, focusing attention to the subgroup of older adults. Importantly, the impact of age on stroke recovery is a topic that has not been studied in depth.

A systematic review shows differences in the care provided and poorer outcomes in older stroke patients compared to younger ones [4]. In our study, the great majority of older patients received speech therapy (90% vs. 44% in younger patients); this could be explained for a greater prevalence of swallowing disorders in the oldest patients. Moreover, the time spent on occupational therapy was slightly shorter in the older adults; this could be explained by the greater fatigability during the performance of activities such as mirror therapy, virtual reality, or somatosensory activities. It is worth noting that other age-related differences such as comorbidities, cardiovascular risk factors, and stroke severity might contribute to this. Nonetheless, contrary to findings from other studies [4,5], no differences in the total time of therapy and length of stay were observed in our study. The mean time of rehabilitation therapies summed more than 170 min per day for at least 5 days per week in all age-groups, close to the 180-min standard that guidelines recommend [7]. This is, in our opinion, the finding that best supports the study feasibility.

In addition to age, many other factors can explain differences on recovery. The interaction of both personal and environmental factors can play a role in clinical outcomes. Clearly, much depends on a patient’s personality and their ability to cope with disease, but the coexistence of comorbidities, caregiving support, access to health facilities, education, and other socioeconomic factors also have a potential impact on patient outcomes, making recovery complex and highly specific. There is a need to determine the most active factors that affect patient outcomes [11].

It is important to evaluate the feasibility of a new intervention in a particular setting [16]. Some of the feasibility issues include whether the intervention can be delivered as planned, the adherence/compliance of participants to treatment, collaboration rates, and loss to follow-up. In particular, compliance with the IRP is associated with significant improvements in the recovery of physical functioning in stroke [18]. Our findings support that an IRP is feasible in stroke patients of all ages fulfilling the above-mentioned criteria.

Some limitations may restrict the generalizability of our results. The patients in this study have been selected to receive an IRP because of their greater potential for recovery. Beside the limitations inherent to rehabilitation samples and the preliminary nature of this study, the exact measurement of time units and patient collaboration is a difficult challenge for the team in charge of therapies. We must also consider that this study was partially conducted during the SARS-Cov2 pandemic where security and health restrictions had an impact not only on the recruitment of patients, but also on the availability of human and material resources.

In our sample, no woman achieved the 180-min standard therapy time, which suggests that women could be a vulnerable group for low intensity. We shall be particularly vigilant in this regard when analyzing the effectiveness of the intervention. Another topic that deserves more attention in further research is how to improve the valuable rehabilitation time during hospital stay. The inclusion of new enriched-environment activities and new rehabilitation methods shall also be approached in further studies.

## 5. Conclusions

We conclude that the IRP is a feasible intervention in patients with subacute stroke, regardless of their age. Older adults have more speech therapy needs, but no differences on the time dedicated to each activity during physical, occupational, and speech therapy were found.

## Figures and Tables

**Figure 1 ijerph-20-04696-f001:**
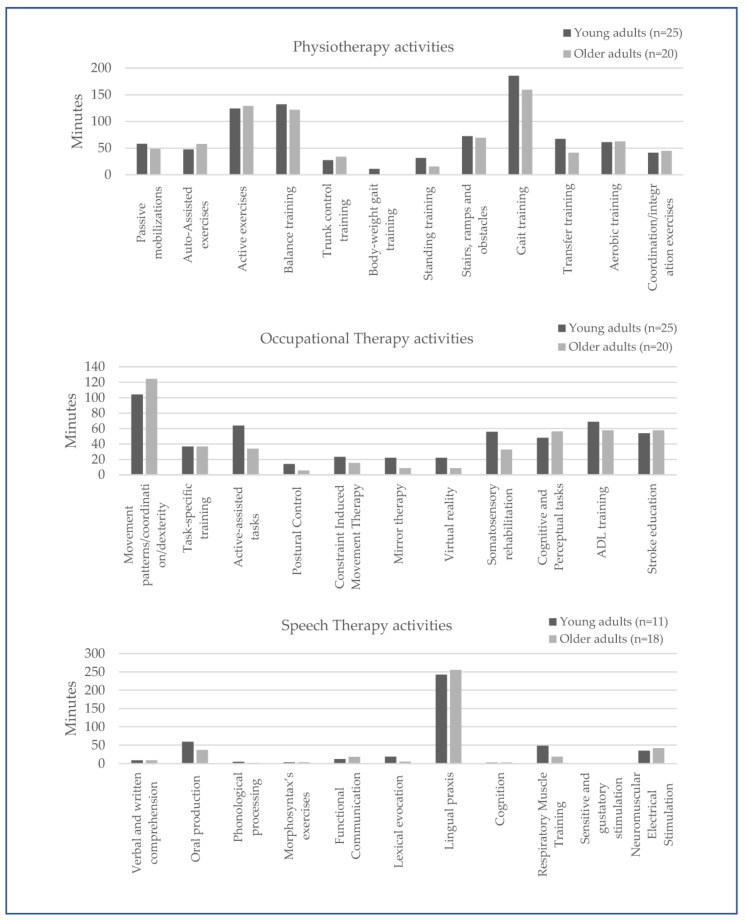
Mean daily time allocated to physical, occupational, and speech therapies in patients according to age.

**Table 1 ijerph-20-04696-t001:** Description of therapies and included therapies during the rehabilitation program.

Rehabilitation Therapies	Activities
Physiotherapy	Passive mobilizations of affected extremities. Auto-assisted exercises of affected extremities. Active exercises of affected extremities. Balance training and stepping. Trunk control training. Body weight supported gait training. Standing training—standing frame. Stairs, ramps, and obstacles. Gait training. Transfer training. Aerobic training. Coordination and integration exercises of the affected side.
Occupational therapy	Upper limb movement patterns, coordination, and dexterity training. Task-specific training. Active-assisted tasks with upper limb weight support. Postural control and orthosis use for affected upper limb. Constraint Induced Movement Therapy (CIMT). Mirror therapy. Virtual reality for upper limb movement training. Somatosensory rehabilitation. Cognitive and perceptual tasks. Activities of daily life training. Stroke education.
Speech therapy	Verbal and written comprehension. Oral production (articulation, volume, tone and prosody). Phonological processing. Morphosyntax exercises. Functional communication. Lexical evocation (free, phonetic phonological, and semantic evocation). Lingual praxis. Cognition (memory, reasoning, logical reasoning, abstraction). Respiratory Muscle Training (Orygen Dual Valve©). Sensitive and gustatory stimulation. Neuromuscular Electrical Stimulation (Vital Stym©).

**Table 2 ijerph-20-04696-t002:** Baseline demographic and clinical characteristics of participants.

	Total Sample (n = 45)	Adults (<65 Years) (n = 25)	Older Adults (≥65 Years) (n = 20)	*p*
Sex, men (%)	35 (77.8%)	21 (84.0%)	14 (70.0%)	0.262
Years of schooling, <8 years, n (%)	22 (38.9%)	8 (30.0%)	14 (70.0%)	0.011 *
Dominance, right-handed, n(%)	44 (97.8%)	24 (96.0%)	20 (100%)	0.556
Smoking history, n (%)				
Never smoked	25 (55.6%)	12 (48%)	13 (65%)	0.129
Former smoker	9 (20.0%)	4 (16%)	5 (20%)	
Current smoker	11 (24.4%)	9 (36%)	2 (10%)	
Alcoholic habits, n (%)	9 (20.0%)	6 (24.0%)	3 (15.0%)	0.710
SDU/day	6.0 (SD 5.3)	5.2 (SD 1.9)	7.7 (SD 9.9)	0.705
Comorbidities, n (%)	37 (82.2%)	17 (68.0%)	20 (100%)	0.005 *
Common comorbidities, n (%)				
Hypertension	28 (62.2%)	12 (48.0%)	16 (80.0%)	0.035 *
Dyslipidemia	18 (40.0%)	7 (28.0%)	11 (55.0%)	0.066
Diabetes mellitus	12 (26.7%)	4 (16.0%)	8 (40.0%)	0.096
Obesity (BMI > 30 Kg/m^2^)	7 (15.6%)	3 (12.0%)	4 (20.0%)	0.682
Ischemic heart disease/heart failure	5 (11.1%)	3 (12.0%)	2 (10.0%)	0.608
Atrial fibrillation	2 (4.4%)	1 (4.0%)	1 (5.0%)	0.697
Sleep apnea	4 (8.9%)	1 (4.0%)	3 (15.0%)	0.224
Stroke onset (days), mean (SD)	6.6 (SD 3.5)	7.1 (SD 3.7)	6.1 (SD 3.2)	0.331
Stroke Type, n (%)				
Ischemic	37 (82.2%)	19 (76.0%)	18 (90.0%)	0.206
Hemorrhagic	8 (17.8%)	6 (24.0%)	2 (10.0%)	
Stroke Hemisphere, n (%)				
Right	16 (35.6%)	8 (32.0%)	8 (40.0%)	0.577
Left	29 (64.4%)	17 (68.0%)	12 (60.0%)	
Oxford classification, n (%)				
Total anterior circulation infarction	9 (20.0%)	5 (20.0%)	4 (20.0%)	0.168
Partial anterior circulation infarction	11 (24.4%)	8 (32.0%)	3 (15.0%)	
Posterior circulation infarction	1 (2.2%)	0 (0%)	1 (5.0%)	
Lacunar anterior circulation infarction	15 (33.3%)	5 (20.0%)	10 (50.0%)	
Unknown data	9 (20.0%)	7 (28.0%)	2 (10.0%)	
Stroke Etiology (TOAST classification), n (%)				
Large-artery atherosclerosis	5 (11.1%)	4 (16.0%)	1 (5.0%)	0.004 *
Cardioembolism	5 (11.1%)	0 (0%)	5 (25.0%)	
Lacunar or small vessel disease	16 (35.6%)	6 (24.0%)	10 (50.0%)	
Undetermined etiology	12 (26.7%)	10 (40.0%)	2 (11.1%)	
Not available in the medical record	7 (15.6%)	5 (20.0%)	2 (10.0%)	
Reperfusion therapy, n (%)	14 (31.1%)	8 (32.0%)	6 (30.0%)	0.885
NIHSS at admission, mean (SD)	6.1 (SD 2.7)	6.0 (SD 3.0)	6.1 (SD 2.5)	0.859
Modified Rankin Scale at admission, mean (SD)	3.3 (SD 0.8)	3.4 (SD 0.8)	3.3 (SD 0.8)	0.268
Barthel Index at admission, mean (SD)	56.9 (SD 17.8)	58.2 (SD 8.5)	55.4 (SD 16.9)	0.609

Abbreviations: SDU: standard drink unit (10 g ethanol); BMI: body mass index; NIHSS: National Institutes of Health Stroke Scale. (*) Statistically significant (*p* < 0.05).

**Table 3 ijerph-20-04696-t003:** Type and duration of therapies performed during stay in the intensive rehabilitation unit.

	Total Sample (n = 45)	Adults (<65 Years) (n = 25)	Older Adults (≥65 Years) (n = 20)	*p*
Length of stay (days), mean (SD)	16.4 (SD 8.9)	16.6 (SD 11.1)	16.2 (SD 5.5)	0.869
Therapy days (days), mean (SD)	13.2 (SD 7.4)	13.5 (SD 9.1)	12.8 (SD 4.5)	0.767
Daily time of therapy (minutes), mean (SD)	173.8 (SD 31.5)	175.1 (SD 28.4)	173.1 (SD 28.9)	0.584
Therapy requirements, n (%)				
Physiotherapy	45 (100%)	25 (100%)	20 (100%)	-
Occupational therapy	45 (100%)	25 (100%)	20 (100%)	-
Speech therapy	29 (64.4%)	11 (44%)	18 (90%)	0.002 *
Speech therapy needs, n (%)				
Aphasia	5 (17.2%)	3 (27.3%)	2 (11.1%)	0.339
Dysphagia	19 (65.6%)	6 (54.5%)	13 (72.2%)	0.432
Others	5 (17.2%)	2 (18.2%)	3 (16.7%)	0.644
Daily time of each therapy (minutes), mean (SD)				
Physiotherapy	85.3 (47.7.2%)	87.7 (46.9%)	82.3 (48.6%)	0.512
Occupational therapy	52.4 (29.3%)	55.7 (29.7%)	48.3 (28.4%)	0.004 *
Speech therapy	40.8 (23.0%)	43.6 (23.3%)	39.2 (23.0%)	0.180

(*) Statistically significant (*p* < 0.05).

**Table 4 ijerph-20-04696-t004:** Characteristics of patients according to the achieved goal of performing at least 180 min of daily rehabilitation therapy.

	Patients Performing <180 min/Day of Therapy (n = 34)	Patients Performing ≥180 min/Day of Therapy (n = 11)	*p*
Age (years), mean (SD)	62.8 (SD 10.7)	63.7 (SD 13.3)	0.820
Sex, women (%)	10 (100%)	0 (0%)	0.041 *
Years of schooling, <8 years, n (%)	16 (72.7%)	6 (27.3%)	0.666
Comorbidities, n (%)	28 (75.7%)	9 (24.3%)	0.968
Stroke onset (days), mean (SD)	6.7 (SD 3.6)	6.5 (SD 3.3)	0.857
Stroke type, ischemic (%)	27 (73.0%)	10 (27.0%)	0.657
Stroke hemisphere, right (%)	23 (78.9%)	6 (20.1%)	0.430
Stroke Location (OXFORD classification), n (%)			0.253
Total anterior circulation infarction	8 (88.9%)	1 (11.1%)	
Partial anterior circulation infarction	8 (72.7%)	3 (27.3%)	
Posterior circulation infarction	0 (0.0%)	1 (100%)	
Lacunar anterior circulation infarction	10 (66.7%)	5 (33.3%)	
Stroke Etiology (TOAST classification), n (%)			
Large-artery atherosclerosis	1 (68.7%)	5 (31.3%)	
Cardioembolism	3 (60.0%)	2 (40.0%)	0.479
Lacunar or small vessel disease	5 (100%)	0 (0%)	
Undetermined etiology	9 (75.0%)	3 (25.0%)	
NIHSS at admission, mean (SD)	6 (SD 2.7)	6.3 (SD 2.9)	0.780
Modified Rankin Scale at admission, mean (SD)	3.3 (SD 0.8)	3.5 (SD 0.5)	0.316
Barthel Index at admission, mean (SD)	58.8 (SD 18.6)	51.0 (SD 13.2)	0.204
Length of stay (days), mean (SD)	14.2 (SD 6.7)	23.7 (SD 11.6)	0.002 *
Therapy days (days), mean (SD)	11.5 (SD 5.6)	18.6 (SD 9.7)	0.004 *

Abbreviations: NIHSS: National Institutes of Health Stroke Scale. (*) Statistically significant (*p* < 0.05).

**Table 5 ijerph-20-04696-t005:** Tolerability and safety outcomes.

	Total Sample (n = 45)	Adults (<65 Years) (n = 25)	Older Adults (≥65 Years) (n = 20)	*p*
**Tolerability variables**				
Lost to follow-up, n (%)	0 (0.0%)	0 (0.0%)	0 (0.0%)	-
Attendance ^1^, mean (SD)	95.8 (SD 5.9)	95.4 (SD 6.0)	96.4 (SD 5.9)	0.632
Reasons for not attending a session, n (%)				
External medical visits or examinations	21 (46.6%)	11 (52.4%)	10 (47.6%)	0.697
Clinical events	3 (6.6%)	3 (100%)	0 (0.0%)	0.083
Need of rest	4 (8.9%)	1 (25.0%)	3 (75.0%)	0.206
Other reasons	4 (8.9%)	3 (75.0%)	1 (25.0%)	0.404
Reasons for early termination, n (%)				
High fatigability	17 (37.7%)	9 (53.0%)	8 (47.0%)	0.783
Poor participation	9 (20.0%)	5 (55.5%)	4 (44.5%)	1.000
Other causes	3 (6.7%)	1 (33.3%)	2 (66.7%)	0.423
Treatment interruption ^2^, n (%)				0.317
External medical visits or examinations	5 (11.1%)	4 (80.0%)	1 (10.0%)	
General discomfort	1 (2.2%)	1 (100%)	0 (0.0%)	
**Safety variables**				
Adverse events occurring during any sessions	0 (0.0%)	0 (0.0%)	0 (0.0%)	1.000

^1^ Attendance: ratio performed/scheduled sessions; ^2^ treatment interruption: missing ≥ three consecutive sessions.

## Data Availability

Not applicable.

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
