# Peer review of "Intensive Rehabilitation Program in Older Adults with Stroke: Therapy Content and Feasibility—Preliminary Results from the BRAIN-CONNECTS Study"

_ijerph, 2023, doi:10.3390/ijerph20064696_

Round 1

Reviewer 1 Report

The work, preliminary to the main project (BRAIN-CONNECTS Project), aimed to evaluate a rehabilitation program for elderly people suffering from a subacute stroke. This is a partial analysis of a cohort of 45 elderly patients after subacute stroke (total project 65 patients) submitted to therapeutic procedures with weekly sessions involving physical, occupational and language activities. The conclusion is favorable to the reliability of the program regardless of age. As this is a preliminary result, the study did not show which type of activity had the greatest impact or whether one of them is more appropriate for elderly people aged under or over 65. The merit of the study lies in the meticulous documentation of the treatment modalities that could be a manual of conduct for these specific patients. Another important detail was to show that there was no difference in its application, without statistically significant differences, in the elderly or very elderly.

Author Response

Thank you very much for your positive insight. The authors appreciate the reviewers’ investment of time and expertise in helping us to improve our manuscript. Given the preliminary nature of this feasibility study, we still have not analyzed the impact of the intervention. We expect to have detailed data on effectiveness of the intervention in a few months (at the end of the project). We would like to highlight the importance of having feasibility studies, especially in the development of complex interventions and multidisciplinary programs. Before testing the efficacy of an intervention through a clinical trial, it is recommended to verify that it can be carried out as proposed in clinical settings. This point has been specifically approached in the Introduction section (lines 79-82). As the reviewer has pointed out, one of the strengths of this study is that provides a detailed description about activities included in an intensive rehabilitation program addressed to stroke patients. The study was aimed to highlight that these therapeutic modalities are applicable in the subgroup of the older patients.

Reviewer 2 Report

Dear authors,

I have studied with great interest the manuscript “Intensive rehabilitation program in older adults with stroke: therapy content and feasibility. Preliminary results from the BRAIN-CONNECTS study”.

The manuscript is clearly exposed and well written. The topic is original. The authors concluded that IRP is a feasible intervention in patients with subacute stroke, regardless of age, and there are no relevant differences on content or duration of therapy. The references are appropriate. The figures correspond to the description in the text, are well designed and reflect important information.

I have some comments:

1.       In the table 1 what Heart disease is meant? Coronary artery disease, congenital heart disease or what? Please, clarify.

2.       Have you performed multivariate analysis to find predictors for the study endpoint?

Generally, I think that this is a very worthy work. I express my gratitude to the authors for their work and my great pleasure in reading their results.

Author Response

The authors appreciate the reviewer’s investment of time and expertise in helping us to improve our manuscript. Thank you very much for your positive insight and the opportunity to clarify the points described below.

REVIEWER COMMENT: In the Table 1 what Heart disease is meant? Coronary artery disease, congenital heart disease or what? Please, clarify.

AUTHORS RESPONSE: In our sample, ‘Heart disease’ refers to coronary heart disease. We have used this more specific term in the revised version.

REVIEWER COMMENT: Have you performed multivariate analysis to find predictors for the study endpoint?

AUTHORS RESPONSE: We appreciate very much the suggestion of the reviewer. Given that this preliminary analysis is aimed to assess feasibility, we intend to complete recruitment to a study, not only effectiveness of the intensive rehabilitation program, but also the factors associated with a good result by performing a multivariate analysis.

Reviewer 3 Report

The manuscript is clearly written and well understood. The contents are clear and interesting. I recommend the publication.

Author Response

The authors appreciate the reviewer’ investment of time and expertise in helping us to improve our manuscript. Thank you very much for your positive insight.

Round 2

Reviewer 2 Report

I think that this is a very worthy work. I express my gratitude to the authors for their work and my great pleasure in reading their results